DeepSpoofNet: a framework for securing UAVs against GPS spoofing attacks

Badar Aziz Ur Rehman 1
Mahmood Danish 1
http://orcid.org/0000-0001-8692-173X Iqbal Adeel 2 adeeliqbal@yu.ac.kr
http://orcid.org/0000-0001-8454-6980 Kim Sung Won 2
http://orcid.org/0000-0001-7005-6489 Akleylek Sedat 3 4 Akleylek@gmail.com
http://orcid.org/0000-0001-6594-8861 Cengiz Korhan 5
http://orcid.org/0000-0002-2133-5286 Nauman Ali 2
1 Computer Science, SZABIST , Islamabad , Pakistan
2 School of Computer Science and Engineering, Yeungnam University , Gyeongsan-si , Republic of Korea
3 Institute of Computer Science, University of Tartu , Tartu , Estonia
4 Department of Computer Engineering, Istinye University , Istanbul , Turkey
5 Department of Electrical Engineering, Prince Mohammad Bin Fahd University , Al Khobar , Saudi Arabia
Coelho Paulo Jorge
Electronic publication date: 2025 Mar 10
Publication date: 2025
Volume: 11
Electronic Location ID: e2714
Received 2024 Apr 27; Accepted 2025 Jan 27
Copyright: © 2025 Badar et al.
Copyright year: 2025
Copyright holder: Badar et al.
License: This is an open access article distributed under the terms of the Creative Commons Attribution License, which permits unrestricted use, distribution, reproduction and adaptation in any medium and for any purpose provided that it is properly attributed. For attribution, the original author(s), title, publication source (PeerJ Computer Science) and either DOI or URL of the article must be cited.
License URL: https://creativecommons.org/licenses/by/4.0/

Keywords: Security, Neural network

Funding: Basic Science Research Program through the National Research Foundation of Korea Ministry of Education NRF-2021R1A6A1A03039493 Korea Government (MSIT) NRF-2022R1A2C1004401 This research was supported by the Basic Science Research Program through the National Research Foundation of Korea (NRF) funded by the Ministry of Education (NRF-2021R1A6A1A03039493) and the NRF grant funded by the Korea Government (MSIT) (NRF-2022R1A2C1004401). No additional external funding was received for this study. The funders had no role in study design, data collection and analysis, decision to publish, or preparation of the manuscript.

==============================
Uncrewed Aerial Vehicles (UAVs) are frequently utilized in several domains such as transportation, distribution, monitoring, and aviation. A significant security vulnerability is the Global Positioning System (GPS) Spoofing attack, wherein the assailant deceives the GPS receiver by transmitting counterfeit signals, thereby gaining control of the UAV. This can result in the UAV being captured or, in certain instances, destroyed. Numerous strategies have been presented to identify counterfeit GPS signals. Although there have been notable advancements in machine learning (ML) for detecting GPS spoofing attacks, there are still challenges and limitations in the current state-of-the-art research. These include imbalanced datasets, sub-optimal feature selection, and the accuracy of attack detection in resource-constrained environments. The proposed framework investigates the optimal pairing of feature selection (FS) methodologies and deep learning techniques for detecting GPS spoofing attacks on UAVs. The primary objective of this study is to address the challenges associated with detecting GPS spoofing attempts in UAVs. The study focuses on tackling the issue of imbalanced datasets by implementing rigorous oversampling techniques. To do this, a comprehensive approach is proposed that combines advanced feature selection techniques with powerful neural network (NN) architectures. The selected attributes from this process are then transmitted to the succeeding tiers of a hybrid NN, which integrates convolutional neural network (CNN) and bidirectional long short-term memory (BiLSTM) components. The Analysis of Variance (ANOVA) + CNN-BiLSTM hybrid model demonstrates superior performance, producing exceptional results with a precision of 98.84%, accuracy of 99.25%, F1 score of 99.26%, and recall of 99.69%. The proposed hybrid model for detecting GPS spoofing attacks exhibits significant improvements in terms of prediction accuracy, true positive and false positive rates, as well as F1 score and recall values.

Introduction

UAV are uncrewed aerial vehicles that operate remotely without the need for human pilots. Equipped with sensitive equipment such as cameras and sensors, these devices serve various purposes including delivery, mapping, inspection, and surveillance. Particularly valuable in emergency, combat, and disaster situations, they can undertake tasks considered hazardous or challenging for humans. Despite their autonomy, UAV require support for operation, presenting cybersecurity challenges such as confidentiality, integrity, and availability (CIA) issues. Similar to traditional aircraft, UAV are classified based on various factors such as flying modes, engine types, ranges, roles, design, and weight (Baig, Syed & Mohammad, 2022). Ground control units allow operators to remotely control UAV, offering services like weather tracking, aerial photography, cargo transportation, and surveillance. Notable examples include the S-100 Camcopter, designed for delivering defense payloads to remote and inaccessible locations. The commercial drone market is poised for significant growth, with forecasts projecting a market share of USD 58.4 billion by 2026.

GPS spoofing, command injection, denial of service (DOS), jamming, and other techniques can all leave modern UAV vulnerable to attacks. Particularly concerning are GPS spoofing attacks because of their ease of use and low cost. Early identification is key to stopping these kinds of attacks. Different tactics and methods are used to thwart GPS and DOS attacks. A GPS assault aims to fool a UAV’s detection capabilities by manipulating its signals, causing confusion between bogus and real signals. These techniques can identify GPS signals; however, they are not sufficient to identify DOS attacks.

In Fig. 1, for instance, a UAV is vulnerable to many threats from the CIA triad. A complete strategy that includes secure communication protocols, robust encryption, strict access restrictions, firmware and software security measures, and physical safeguards is required to protect UAV against such assaults.

Figure 1 Types of attacks on UAV in CIA triad.

Big tech companies like Facebook and Tesla use UAV to provide internet connectivity to rural areas lacking access. UAV excel in tasks classified as tedious, dirty, and dangerous; indicating situations beyond human capabilities to operate and intervene. UAV will rapidly dispatch disaster alerts and assist in expediting rescue or recovery operations when a public communication network is overwhelmed. They can transport medical supplies to locations that are difficult to get to. UAV can effectively cover vast areas while maintaining human safety, particularly in scenarios involving the detection of hazardous gas leaks, wildfires, or wildlife monitoring.

The most widespread and important use of UAV is for military purposes. Superior UAS have a noticeable advantage over their competitors due to their reduced size, enhanced stealth, and ability to operate in difficult environments and monitor borders. UAV can effectively combat terrorism in various situations without causing any casualties. To monitor in real-time, the UAV needs to gather a lot of precise information during its flight. These data comprise commands from the base station to control the UAV and guarantee mission performance, as well as a payload subsystem and telemetric subsystem (Khan et al., 2022).

Communication lines, sensors, and software should be the primary focus of studying UAV vulnerabilities. Unlike the sophisticated communication technologies used in military UAV, consumer-grade UAV often rely on direct communication techniques such as the Wireless Fidelity (Wi-Fi) b/g/n standard and the C-band electromagnetic spectrum (Qiao, Zhang & Du, 2017). Mobile GCS typically operate these small UAV. For Da-Jiang Innovations (DJI) UAV and many others, the ground control station (GCS) usually consists of remote controllers and smartphones. Communication channels are susceptible to disconnection, jamming, eavesdropping, and man-in-the-middle attacks. Sensor vulnerabilities include the risk of GPS spoofing compromising the devices (Meng et al., 2020) and the potential for sound disturbances to interfere with gyroscopic sensors. While ensuring swift recovery and resumption of normal operations following an attack is essential, identifying potential attacks is even more critical. While GPS spoofing may be challenging to detect, jamming and disconnection attacks are easily recognizable.

GPS spoofing attacks become more sophisticated with advancements in location-based services, posing a threat to the reliability of transportation, timing, and navigation applications. Current IDS face challenges due to imbalanced datasets, resulting in decreased GPS spoofing detection performance. Furthermore, distinguishing between legitimate and fake GPS signals using traditional methods proves inadequate, exacerbated by reliance on expensive specialized equipment or central infrastructure. Consequently, machine learning models trained with imbalanced datasets may misclassify fake GPS signals as genuine, resulting in higher false negative rates and poor deep learning model performance.

This study aims to address these challenges by improving the detection accuracy of GPS spoofing attacks, minimizing false positive rates through the identification of optimal feature selection methods, and addressing imbalanced datasets. The proposed method focuses on detecting attacks on UAV efficiently, using minimal resources, and maximizing readability. Research objectives include developing an effective and adaptable method for detecting attacks, ensuring timely detection, and enhancing readability using easily available sensor data. Research questions helped the study find ways to improve the current state of the art, fix the problem of unbalanced datasets for best model training, find the best engineering features for model training, and figure out the most important performance analysis parameters for checking how well the proposed model works. This study contributes to the field by enhancing GPS spoofing detection accuracy for UAVs through an adaptable method that improves detection reliability. It minimizes false positives by leveraging optimal feature selection techniques and addresses the issue of imbalanced datasets for more robust model training. The proposed method efficiently utilizes readily available sensor data, ensuring timely detection with minimal resource consumption. Moreover, it provides a comprehensive performance analysis, identifying critical metrics to assess the model’s effectiveness, scalability, and adaptability in diverse UAV operational scenarios.

The rest of the article is organized as follows: the “Literature Review” section is a literature review that comprehensively analyzes the existing state of the art regarding GPS spoofing attacks on UAV, highlighting the limitations of current methods. The “Proposed Model” section is the proposed method that addresses imbalanced datasets and feature selection for detecting GPS spoofing attacks on UAV. In the “Results and Discussion,” details the steps taken to develop and implement the proposed method, including data collection, model training, and evaluation. The “Comparative Analysis” section presents the findings of the study, including the performance of the proposed model in detecting GPS spoofing attacks as well as a comparison with existing methods. Finally, the “Conclusion and Future Work” section summarizes the study’s key findings, discusses the implications of the results, and suggests directions for future research.

Literature review

Recent research on UAV security has generated significant attention due to the rise in popularity of drones and associated security issues. Considerable resources have been dedicated to creating algorithms for detecting GPS spoofing. Media stories are focusing more on cyber-attacks against UAV, which are now seen as a drawback to their beneficial utilization. The International Telecommunication Union (ITU) is contemplating integrating UAV into its Fifth Generation (5G) wireless communication framework. Furthermore, following the earthquake and tsunami that struck Fukushima, Japan in 2011, A Honeywell T-Hawk drone was used to examine a nuclear plant that was too radioactive for humans to enter (Xue et al., 2020). In 2012, someone interfered with the GPS signal of a rotor-assisted UAV called Camcopter S-100, causing it to crash fatally into a ground control operations center. This unforeseen disturbance led to the sudden and unnecessary death of an engineer, as well as injuries to both remote pilots (Pocock, 2012).

GPS spoofing attack detection using non-ML methods

GPS spoofing is a security risk that can trick the receiver into computing inaccurate position data. Various techniques, such as non-ML as shown in Table 1 or conventional methods, are used to detect GPS spoofing in UAV. Non-ML methods include signal strength analysis, timing analysis, signal quality analysis, consistency checks, and cryptography techniques.

Table 1 Exploring non-ML approaches for detecting GPS spoofing attacks.

Reference	Technique	Dataset	Evaluation parameters	Results	Limitations	
GNSS, Spoofing Detection/GPS Attack (Rothmaier et al., 2021)	GLRT	TEXBAT	RSP, auto-correlation function, distortion & pseudo-range residuals	Experiment FA-30s, 0.4 dB power advantage	Unexpected phenomena affecting metric behavior: Aircraft rotation during takeoff and receiver clock resets.

Impact of over-bounding error models: These models captured the effects but reduced detection performance.

	
Small UAVs, GPS Attack (Basan et al., 2022)	KLD	Simulated dataset	UAV flight altitude, Number of satellites, GPS speed, Flight angle, Latitude Longitude	Experiment (Soft mode simulation) Type II Error = 0.01 Attack Detection Probability 0.99	KL divergence properties: is non-negative, asymmetric, and does not satisfy the triangular inequality.

	
GNSS Spoofing Detection GPS Attack (Magiera, 2019)	Multi-antenna Scheme	Simulated dataset	Received signals, autocorrelation function, distortion, and pseudo-range residuals	Experiment C/N0-45 dB Hz, Pd-99.9%	Hardware trade-off: A trade-off exists between hardware complexity and effectiveness.

Antenna efficiency: More antennas can improve spoofing detection in low carrier-to-noise ratio conditions.

	
Civilian UAVs GPS Attack Detection Using Visual Odometry (Varshosaz et al., 2019)	SEDCP	Simulated dataset	Windows size, detection rate	SEDCP (w = 21), 32 (100%), (HOD_AD) (w = 21), 27 (84%), (HOD_TD), (w = 21), 29 (91%)	Effectiveness limitations: Velocity changes compromise VO in detecting spoofed UAV trajectories.

Performance impact: Velocity changes reduce image overlap, hindering performance.

	
Analyzing Camera’s Video Stream to detect GPS Attack (Davidovich, Nassi & Elovici, 2022)	VISAS	Simulated dataset	Windows size, altitude, maximum error distance	Altitude-50 m, window size (4), maximum error distance-5 m, lowest error rate-4 m	Detection limitations: Night and low-texture areas (e.g., water or snow).

	

Rothmaier et al. (2021) introduce a generic framework that uses the Generalized Likelihood Ratio Test (GLRT) to merge many measurements for Global Navigation Satellite System (GNSS) spoofing detection. The framework was created to be resilient against various attack modes while also keeping a low rate of false alerts, according to the Neyman-Pearson paradigm. The study provides an overview of the GLRT framework previously published and evaluates its performance against traditional logical gate-based approaches for combining measurements, received signal power (RSP), and signal distortion metrics. The results indicate a 60% decrease in the maximum probability of miss-detection (PMD) scenario. The authors modify measurement models for a receiver model using flight data and confirm adherence to the false alarm (FA) guarantee for power, signal distortion metric, and pseudo-range residuals. The authors proved the framework’s usefulness by utilizing it in spoofing scenarios from the TEXBAT dataset with the same receiver type and combining three measures to achieve strong detection performance.

Basan et al. (2022) introduced a GPS spoofing detection technique for UAV that uses mathematical tools to address the issue without the need for extensive training data. In detecting GPS spoofing attacks in a group of UAV, a proposed method achieves 96% accuracy in detecting attacks, with a 3.5% rate of false positives. The text discusses key aspects of data analysis and normalization procedures for data analysis, as well as the use of the Kullback Leibler Divergence (KLD) measure to identify abnormalities in UAV systems. The article offers a method for detecting and minimizing the effects of intermediate GNSS spoofing, which involves transmitting bogus signals with tiny temporal differences from genuine signals received from satellites. The suggested anti-spoofing system combines techniques for antenna array processing with a multi-path identification algorithm to tell the difference between real and fake GNSS signals that are very similar to each other. The spoofing detection mechanism compares the steering vectors of received spatial components. The mitigation strategy uses adaptive beamforming to eliminate interference from shared directions and maintain original signals from GNSS satellites. Simulations confirm the efficacy of this approach in protecting GNSS receivers against intermediate spoofing interference, showcasing its usefulness. The authors in Varshosaz et al. (2019)’s proposed method compares the UAV’s relative sub-trajectory from visual odometry (VO) with its absolute replica from GPS, utilizing dissimilarity measures sum of Euclidean distances between corresponding points (SEDCP), angle distance, and taxicab distance. This allows for precise identification of UAV spoofing events, limiting the occurrence of VO drift errors. The approach demonstrates wide applicability, real-time execution, and efficiency in identifying different UAV spoofing cases. The results show that the method is effective in identifying instances of UAV spoofing, especially during long-range UAV flights with large changes in flight direction greater than 3 degrees and gradual UAV spoofing scenarios with a redirection rate of 1 degree. SEDCP is highly effective at identifying spoofing without redirection, providing excellent detection of false GPS locations. This solution dramatically enhances UAV security by accurately detecting spoofing and mitigating VO drift errors.

Mykytyn et al. (2023) aims to develop a system that can identify both single-transmitter and multi-transmitter GPS spoofing attempts to reduce their negative impacts. The detection mechanism identifies and confirms GPS spoofing attacks by comparing the distances between swarm members using GPS coordinates and Impulse Radio Ultra-Wideband ranging and determining if the discrepancy exceeds a predefined threshold. In Davidovich, Nassi & Elovici (2022) the authors suggest extracting frames from the video stream along with their respective GPS coordinates. The approach can effectively detect possible GPS spoofing attacks on a drone by evaluating the correlation between each frame. Operating a drone at an altitude of 50–100 m over an urban area, moving at an average speed of 4 km/h, and in low-light conditions, this approach demonstrates a high level of security. The system can identify GPS spoofing attacks, recognizing when the falsified location deviates by −4 m, with an average variance of 2.5 m from the actual location.

GPS spoofing attack detection using ML methods

ML methods are superior for identifying GPS-spoofing assaults on UAV. GPS systems play a crucial role in navigation, transportation, and essential infrastructure, which makes them susceptible to malevolent individuals. These malicious individuals aim to mislead recipients by sending counterfeit signals, which might result in malicious individuals circumventing conventional anti-spoofing techniques due to their limitations. Analyzing complex patterns in the data using ML techniques can accurately identify genuine and fake GPS signals. The models may learn from large datasets to identify tiny anomalies and variations indicating a spoofing attack. Tables 2–4 summarize the literature discussed in this section.

Table 2 Exploring ML approaches for detecting GPS spoofing attacks.

Reference	Technique	Dataset	Evaluation parameters	Results	Limitations	
Comparison of ROS, RUS using SMOTE, (Wongvorachan, He & Bulut, 2023)	SMOTE, SMOTE-NC	High School Study HSLS: 09 dataset	Accuracy (ACC), Precision, Recall, ROC-AUC, F1-score	SMOTE-NC + RUS ACC-0.905%, Rec-0.898%, F1-0.904%, Pre-0.911%, ROC-AUC-0.967%	∙ Study constraints: Limited to a single dataset and challenges in determining optimal hyper-parameters.	
Impact of Dataset and Parameters on ML Model, (Talaei Khoei et al., 2023)	SVM, ANN, RF, GNB, CART, LR	Generated dataset (Aissou et al., 2021)	ACC, Probability of FA, Probability of detection, PMD	GNB ACC-91.2%, PD-86.16%, PFA-2.23%, PMD-13.84%, RF ACC-99.43%, PD-99.6%, PFA-1.01%, PMD-1.8% CART ACC-99.9%, PD-99.98%, PFA-1.005%, PMD-0.02% LR ACC-91.2%, PD-86.19%, PFA-3.0%, PMD-13.84% ANN ACC-93%, PD-93.4%, PFA-3.37%, PMD-6.6%	Generalization: needs to be assessed on independent test sets.

Imbalanced datasets: may affect performance.

Limited diversity: in the dataset can constrain effectiveness.

Additional metrics: beyond accuracy should be evaluated.

	
FS Method for IoT IDS using ML, (Albulayhi et al., 2022)	IG, GR, UMF, IMF	IoTID20 and NSL-KDD datasets	ACC, Precision, Recall, F1-Measure	UMF ACC-99.98%, PR-99.90%, Rec-99.90%, F1-99.90% IMF ACC-99.98%, PR-99.90%, Rec-99.90%, F1-99.90%	Computational expense: Random Forest for feature selection is computationally intensive and struggles with high-dimensional datasets, leading to model overfitting.

	
Effective Class Imbalance Learning using SMOTE & CNN, (Joloudari et al., 2023)	acpDNN, acpCNN, SMOTE	KEEL dataset (Derrac et al., 2015), breast cancer, Z-Alizadeh Sani dataset	ACC, Performance, Specificity, Precision, Recall, F1-Measure	SMOTE-NORM-CNN ACC-98.57%, Rec-98.58%, F1-98.57%, Pre-98.58%, Spe-98.42%, AUC-99.14%	Training time and computational costs: Relative to traditional ML techniques.

	
Hybrid Noise Handling Technique (Data Balancing & FS), (Puri & Kumar Gupta, 2022)	SMOTE, SMOTE–ENN, K-Means-SMOTE	Glass1, Glass0, Ecoli1, Glass0123vs456, Ecoli2, Glass6, Ecoli3, Yeast-2vs4, Glass016vs2, Yeast1vs7, Yeast1458vs7 datasets	ACC, PMD, Probability of FA	K Means SMOTE Hybrid-1.2%, Bagging-2%, Boosting-2.7%, K Means SMOTE ENN Hybrid-1.2%, Bagging-1.9%, Boosting-2.8%, Without resampling Hybrid-1%, Bagging-2.6%, Boosting-2.3%, SMOTE Hybrid-1.5%, Bagging-1.9%, Boosting-2.5%, SMOTE ENN Hybrid-1.2%, Bagging-1.8%, Boosting-2.8%	Model comparison: A comparison of hybrid SMOTE, K-Mean-SMOTE, and SMOTE-ENN using 11 datasets reveals that the proposed K-Mean-SMOTE-ENN hybrid model is effective only when noise is limited to 20%.

	
ML based IDS (Hybrid FS), (Yin et al., 2023)	IG, RF	UNSW-NB15 dataset	ACC, Precision, Recall, F1-score	IG &RF Uni ACC-80.60%, F1-80.36%, Per-83.30%, Rec-80.60% IG &RF Inter ACC-82.90%, F1-81.67%, Per-81.84%, Rec-82.90% IGRF-RFE ACC-84.24%, F1-82.85%, Per-83.60%, Rec-84.24%	Performance metrics: This study demonstrates low accuracy, precision, recall, and F1 score; the proposed hybrid FS approach exhibits worst-case computational complexity.

	
Small UAV, CBA, GPS Attack, (Wu et al., 2023)	BiLSTM, SHAP, NN	Simulations dataset	Detection time, ACC	CBA parameters: 1,956, Detection time: 367 us, GPS Spoof attack: 99.1%	Feature selection: is not optimal for GPS spoofing attack detection.

Under-sampling: is employed, adversely affecting model training.

Increased complexity: The use of multiple ML methods adds to the complexity of the model.

	
Single Frequency Receiver (GPS Attack) (Shafiee, Mosavi & Moazedi, 2018)	KNN, NN	Simulated dataset	Detection time, complexity	NN structure (proposed) Detection time 2.895 s, Complexity-26, Naive Bayesian DT-0.153, Complexity-N/A	Higher complexity: can be improved by employing more efficient machine learning methods.

	
GPS spoofing attack detection using Perception Data/GPS Attack (Wei, Wang & Sun, 2022)	Perception Data Framework Based on XGBoost model, RF model	Real Flights dataset	ACC, Precision, Recall	PERDET (RF model) ACC-99.69, Precision-99.07, Rec-99.38, F1-Measure-99.22	Biased dataset: is utilized for training, with a 4:1 ratio between normal and attacked data.

	

Table 3 Exploring ML approaches for detecting GPS spoofing attacks-Continued.

Reference	Technique	Dataset	Evaluation parameters	Results	Limitations	
GPS spoofing attack detection in UAS (GPS Attack) (Aissou et al., 2022)	SVM, KNN	Simulated dataset	ACC, PD, Processing Time, Memory Size, and Detection Time per sample	Nu-SVM Processing time (s)-0.12, Memory size (MiB)-0.508, TPR-91.26%, FNR-8.73%, FPR-6.02%, ACC-92.78%	Miss-detection rate: of the proposed model is higher and overall average results that can be improved using more efficient methods.

	
GPS in Vehicles (GPS Attack) (Jiang, Wu & Xin, 2022)	Recurrent Neural Network (RNN)	Real-time Data, BDD-100K (Yu et al., 2020)	Detection Rate, ACC	ACC-80%, FA-7.9%, DR-90.3%	RNN limitations: to detect GPS spoofing attacks due to a lack of contextual information and their inability to capture long-term dependencies in GPS data sequences.

	
Datasets Effectiveness compare/GPS Attack (Jullian et al., 2022)	Deep Neural Networks (DNN)	MAVLINK, TEXBAT datasets	ACC, Precision, Recall, F1-Score	MLP (MAVLINK), F1 (92.29%), ACC (99.93%), MLP (MAVLINK), F1 (97.41%), ACC (94.43%), MLP (TEXBAT), F1-score (82.79%), ACC (83.23%)	Resource management challenges: Handling UAVs’ computational, memory, and storage requirements becomes challenging as the model’s complexity, including the number of units, hidden layers, and features, continues to increase.

	
UAS/GPS Attack (Manesh et al., 2019)	NN	Software Defined Radio (SDR) based dataset	Pseudo-range, Doppler shift, SNR	ACC-98.3%, DR-99.2%, FA-2.6%	Feature reliance: The proposed model heavily relies on pseudo-range, Doppler shift, and SNR for classification, potentially overlooking important characteristics such as signal structure, timing information, and trajectory data of GPS spoofing signals.

	
UAVs/GPS Attack (Wang et al., 2020)	NN LSTM	MATLAB Simulations dataset	Detection Rate Detection Time	DT-3 s, DR-78%	Model performance: Based on the results, the proposed model demonstrates poor performance with a low detection rate for GPS spoofing attack detection.

	
IDS (light-weight)/GPS Attack, low energy (Arthur, 2019)	Multi-class SVM, Deep-Q Network	Simulations dataset	ACC	Multi-class-SVM, ACC-78%, SLT + Multi-class SVM, ACC-94%	Dataset size: is very low, increasing the risk of overfitting.

Inefficient feature selection: used in this model.

	
Hierarchical Detection and Response System Cyber Attacks in UAVs (Sedjelmaci, Senouci & Ansari, 2018)	SVM	Simulations dataset	Detection Rate Efficiency	DR-93%, FPR-3%	Feature relevance: SSI, NPS, JITTER, and NPD have limited relevance and discriminatory power.

Need for additional features: like carrier-to-noise ratio and timing anomalies should be explored to improve detection accuracy.

	
Small UAVs, Deep Learning Based/GPS Attack (Sun et al., 2023)	PCA, CNN, LSTM	GPS signal dataset	ACC, Precision, Recall Error, F1 Score	PCA-CNN-LSTM, ACC-0.9943%, Perc-0.9798%, Rec-0.965%, F1-0.9722%	Model complexity and computational performance: of the GPS signal spoofing detection model can be improved.

Feature selection: is not prioritized in this study for optimization.

	

Table 4 Exploring ML approaches for detecting GPS spoofing attacks-continued.

Reference	Technique	Dataset	Evaluation parameters	Results	Limitations	
IDS for IoD (Escorcia-Gutierrez et al., 2023)	DBN, SSO, STFA-HDLID (Hybrid Model)	Simulated dataset	ACC, Sensitivity	STFA-HDLID ACC-98.85%, Sensitivity-99.36%, LSTM-RNN, ACC-98.02%, Sensitivity-97.44%, SVM ACC-97.46%, Sensitivity-97.21%	DBN challenges: DBN faces challenges in training complexity and computational requirements.

SSO algorithm limitations: in convergence speed and robustness in noisy or dynamic UAV environments.

	
Satellite imagery matching approach/GPS Attack (Xue et al., 2020)	DeepSIM, DNN, ResNet	SatUAV dataset	ACC, Precision, Recall Error, F1 Score	On-ground ACC-0.948%, PER-0.930%, Rec-0.979%, Error-0.052%, F1-0.954 % On-board ACC-0.890%, PER-0.871%, Re-0.936%, Error-0.110%, F1-0.903 %	Inefficient feature selection: used i.

Detection complexity: Detecting attacks at night, in wind, and bad weather is complex and suboptimal.

	
Hybrid IDS for feature selection/GPS Attack (Liu & Shi, 2022)	RF, GA-RF	UNSW-NB15, NSL-KDD (Derrac et al., 2021) datasets	ACC, False Positive Rate	RF based IDS ACC- 94.7%, FPR-2%, Dataset-KDD’99, GA-RF (Proposed) ACC-96.12%, FPR-2.91%, Dataset-NSL-KDD,GA-RF (Proposed), ACC- 92.06%, FPR-1.60%, Dataset-UNSW-NB15,	Random forest limitations: Random Forest for feature selection is computationally expensive and struggles with high-dimensional datasets.

	
CONSTDET: Control Semantics-Based Detection/GPS Attack (Wei et al., 2022)	CONSTDET	Realtime dataset	ACC, Precision, Recall, Missing, Mistake, F1 Score	CONSTDET ACC-97.70%, Perc-98.70%, Rec-96.76%, F1-97.72%, Missing-3.24%, Mistake-1.32%	PID controllers importance: PID controllers are crucial for UAV, ensuring stable flight.

	
Comparative Analysis of the Ensemble Models for Detecting/GPS Attack (Gasimova, Khoei & Kaabouch, 2022)	Bagging, Stacking, Boosting	Simulated datasets	ACC, PMD, Probability of FA	Stacking ACC-95.43%, Pd-99.56%, PMD-0.36%, PFA-0.03 Bagging ACC-95.28%, Pd-99.24%, PMD-0.64%, PFA-1.07% Boosting ACC-94.61%, Pd-96.55%, PMD-2.95%, PFA-5.08%	Ensemble learning limitations: Ensemble-based machine learning techniques increase computational complexity and make interpreting the decision-making process difficult.

	
UAV IDS Testing/GPS Attack (Basan, 2022)	Chi-square distribution, Pearson Correlation Coefficient (PCC)	Simulations dataset, Real-time dataset	Probability, Standard deviation	GPS Uncertainty 13:10 and 13:20	Anomalous data generation: The generated anomalous data is close to real data, but no ML model is proposed for data generation.

	

Using the 2009 High School Longitudinal Study dataset, Wongvorachan, He & Bulut (2023) explored several sampling approaches to address class imbalance in diverse circumstances. This study compares random oversampling (ROS), random undersampling (RUS), and a hybrid re-sampling method that combines Synthetic Minority Oversampling Technique for Nominal and Continuous (SMOTE-NC) with RUS. To evaluate each method, classification uses a random forest (RF). For moderately imbalanced data, ROS works, but hybrid re-sampling works better for very skewed data. The authors also proposed future research and discussed how these findings may affect educational data mining. The authors in Talaei Khoei et al. (2023) compared artificial neural networks (ANN), classification, regression decision tree (CART), logistic regression (LR), GaussianNB (GNB), RF, and support vector machine (SVM). Measurements included hyperparameters, dataset size, correlated features, imbalanced datasets, and regularization for thirteen distinct aspects of actual and manufactured GPS assault signals were investigated. Four criteria graded the models: PMD, accuracy, probability of false alert (PFA), and probability of detection (PD).

Albulayhi et al. (2022) developed a novel method for selecting and extracting features from anomaly-based intrusion detection system (IDS) for Internet of Things (IoT). This method extracted significant features in different proportions using information gain (IG) and gain ratio (GR) entropy-based methods. Mathematical set theory extracts the most valuable properties, especially using union and intersection procedures. Researchers used the IoT Intrusion Dataset 2020 (IoTID20) and Network-based Survivable Learning-Knowledge Discovery (NSL-KDD) datasets to train and evaluate the model framework to determine its efficacy. The authors used bagging, multilayer perceptron (MLP), J48, and instance-based learning with k nearest neighbors (IBk) in this experiment. Using the intersection operation, 11 of the 86 IoTID20 features and 15 of the 41 NSL-KDD features were the union operation resulting in 28 IoTID20 and 25 acNSL-KDD characteristics. This method excelled with 99.98% classification accuracy. Alomari et al. (2023) proposed a novel malware detection approach using deep learning and feature selection. Researchers used two datasets—one with malware and one with innocent activities—to create a training system. The authors created multi-feature-selected datasets using pre-processing and correlation-based feature selection. The authors trained dense and long-short-term memory networks (LSTM) based deep learning models on these feature-selected datasets. The authors measure model performance using precision, recall, accuracy, and the F1-score. An intriguing finding was that certain feature-selected situations performed similarly to the baseline dataset. Performance decreases vary by dataset. Performance degradation in the first dataset ranges from 0.07% to 5.84%, with feature reduction ratios between 18.18% and 42.42%. The second dataset shows an 81.77–93.5% decrease rate and 3.79–9.44% performance degradation.

The authors in Puri & Kumar Gupta (2022) propose a novel re-sampling method for imbalanced datasets. The noise-reduction method uses K-means Synthetic Minority Oversampling Technique (SMOTE) oversampling to resolve this issue and the datasets are initially clustered using K-means. SMOTE constructs synthetic minority class instances in clusters to resolve the class imbalance. Finally, it eliminates noise with edited nearest neighbor (ENN), noise is eliminated. Performance tests were done on 11 binary, unbalanced datasets using area under receiver operating curve (AUC). The results showed that the proposed method outperformed others in terms of varying attributive noise. The proposed method also performed well on binary imbalanced datasets with substantial attribute noise. The authors in Yin et al. (2023) combined RF with IG filters which improves feature selection. This strategy uses RF to manage the influence of less significant features chosen by IG based on their high-frequency values. Thus, the feature subset search space has more relevant features. Recursive feature elimination (RFE), a ML-based wrapper method, reduces feature dimensions by assessing the importance of related features in the second phase. Experimental results on the University of New South Wales—Network-Based 2015 (UNSW-NB15) dataset confirm the suggested method’s anomaly detection accuracy improvement. As shown, MLP multi-classification accuracy improves from 82.25% to 84.24%, while the number of features decreases from 42 to 23. The authors in Joloudari et al. (2023) examine how DNN and CNN work with other methods to handle imbalanced data. Oversampling, undersampling, and CNN-based SMOTE integration are involved. These methods are tested utilizing the KEEL, breast cancer, and Z-Alizadeh Sani datasets. Repeat the trials 100 times with random data distributions to ensure reliability. The SMOTE-Normalization-CNN model outperformed other methods with 99.08% accuracy on 24 unbalanced datasets. Thus, this flexible hybrid model effectively addresses imbalanced binary classification difficulties in real-world datasets.

The authors in Wu et al. (2023) propose an interpretability-focused detection methodology CNN-BiLSTM-Attention (CBA) using Shapley Additive Explanations (SHAP). This novel system uses GPS, IMU, and gyroscope data to target UAV. This model aims to address the constraints of standard attack detection systems, which use NN models that complicate detection results and reduce trustworthiness. The framework uses local and global explanations to maintain interpretability. Local explanations reveal the individual causes of detection outcomes by examining each input and model choice. The framework also provides global explanations of the model’s key aspects and how they relate to attack types. SHAP and generally available sensor status data form a framework that provides deeper insight into the relationship between feature values and a variety of attack scenarios, improving UAV attack detection reliability and comprehensibility. The authors in Shafiee, Mosavi & Moazedi (2018) developed a novel assault detection method by studying GPS signal characteristics. They used a multi-layer neural network, an enhanced K-nearest neighbor (KNN) algorithm, and a Bayesian classifier to identify several targets. To extract GPS signal features, the detection method used early-late phase, delta, and signal level to extract GPS signal features. A novel multi-layer neural network algorithm that uses feature selection as inputs to improve GPS spoofing detection. The neural network showed good identification accuracy in short simulations on a software GPS receiver. The authors in Wei, Wang & Sun (2022) propose detecting UAV GPS spoofing using a machine-learning technique and perceptual data. For features, they chose a barometer, GPS, magnetometer, gyroscope, and accelerometer. Despite their many disadvantages, these sensors’ diversity allows them to offset perceptual data and collect experimental data during actual flights, making their proposed PerDet solution more feasible. The study in Aissou et al. (2022) examined five instance-based learning models for detecting GPS spoofing in uncrewed aerial system (UAS). Radius neighbor, KNN, C-SVM, linear SVM, and nu-SVM were these models. The study used software-defined radio modules to capture and initialize satellite signal data, as well as simulated simplistic, intermediate, and complex GPS spoofing assaults. After evaluation, Nu-SVM outperformed other instance learning classifiers in accuracy, detection, FA, and miss-detection. The nu-SVM model showed good memory and processing efficiency during detection.

Jiang, Wu & Xin (2022) established deep learning-based approach for pose estimation (DeepPOSE), a deep learning model. Researchers developed a novel deep-learning model to identify fake GPS signals in mobile systems and noisy sensor data. The authors used convolutional and recurrent neural networks to decrease noise and deliver exact vehicle trajectories based on sensor data. They also developed a novel way to precisely display sensor data on Google Maps, decreasing trajectory determination errors. By reconstructing sensor trajectory data, the suggested method may detect GPS spoofing attacks. Since it detects such attacks more accurately than other methods, this strategy is better. Jullian et al. (2022) used a new GPS spoofing defense using a multi-layer perceptron neural network. GPS signals and flight parameters are inputs, and GPS spoofing attacks trigger an alarm. Note that dataset analysis affects this system’s accuracy. TEXBAT data shows 83.23% accuracy, implying that GPS spoofing assaults occur 83.23% of the time. The system’s MAVLINK dataset GPS spoofing detection accuracy is 99.93%, better than its competitor. The system detects attacks 99.93% of the time. The accuracy scores above only apply to the proposed solution’s evaluation using the TEXBAT and MAVLINK datasets. Manesh et al. (2019) proposed a new GPS spoofing detection method that employs artificial neural networks and supervised ML. Classifying GPS signals using pseudo-range, signal-to-noise ratio (SNR), and Doppler shift is easier. The authors compare two-hidden-layer neural networks with different numbers of hidden neurons. The results show that their machine-learning method detects spoofing signals with few FA. Wang et al. (2020) developed a novel GPS spoofing detection approach using the LSTM algorithm. UAV follow a predetermined flight route and help identify GPS spoofing attempts, improving detection rates. We’re proud to be the first to use ML to detect attacks. Extensive testing has confirmed that our GPS spoofing detection method delivers fast and accurate results. Simulated trials accompany our technique to combat GPS spoofing attacks. The experiments showed that our technique detected UAV GPS spoofing attacks quickly and accurately. No equipment upgrades are necessary. This study thoroughly describes our method.

Arthur (2019) explores that the ground or airborne vehicles within transmission range have the ability to spoof and hack drones. These safeguards are insufficient. Attackers can take control of the drone’s flight or autopilot. Due to intermittent network connectivity, drone communication is complicated. To solve these issues and safely return the drone, there is a need for an adaptive IDS that leverages deep learning. Arthur (2019) suggested an IDS that can detect threats in unknown environments using a multiclass SVM and self-taught learning (STL). The Deep-Q Network’s dynamic route learning allows the IDS to self-heal and guide the drone home. These technologies give the IDS a high true positive rate. Simulation findings show that the proposed IDS can defend drones against cyber security assaults due to its accuracy, sensitivity, and specificity. Cyber attacks are typical, suspicious, malicious, or aberrant based on their type. The worst cyber attacks on UAV networks include gray hole and black hole attacks, jamming, GPS spoofing, and misleading information dissemination. Sedjelmaci, Senouci & Ansari (2018) conducted numerous simulations and demonstrated that their proposed approach can detect these attacks from multiple UAV and attackers with low communication costs. They trained the model with Signal Strength Intensity (SSI), Number of Packets Sent (NPS), Jitters, and Number of Packets Dropped (NPD). High detection and low false-positive rates characterize this model (Sedjelmaci, Senouci & Ansari, 2018).

Sun et al. (2023) uses deep learning to detect GPS signal spoofing in small UAV. The authors talk about the UAV hardware system, the experiment jammer, the collection settings such as time and weather, Spearman correlation coefficients for preprocessing, and SVM-SMOTE’s data imbalance solution for getting the GPS signal dataset and preprocessing it. The new principal component analysis (PCA)-CNN-LSTM approach extracts features, CNN captures local features, and LSTM processes and models. Tenfold cross-validation ensures the robustness of the simulation experiment and compares the model to ML and deep learning methods. The PCA-CNN-LSTM neural network model achieves a very high accuracy value which is 0.9949%. This study lays the groundwork for micro UAV GPS signal spoofing detection. Escorcia-Gutierrez et al. (2023) created the Sea Turtle Foraging Algorithm with Hybrid Deep Learning-based Intrusion Detection (STFA-HDLID). This method identifies and classifies Internet of Drones (IoD) intrusions and for accuracy, the authors used min-max normalization and Sea Turtle Foraging Algorithm (STFA) feature selection. The authors also categorized them using Deep Belief Network (DBN) and Sparrow Search Optimization (SSO) and proposed a new IoD intrusion detection mechanism. The authors extensively tested STFA-HDLID on a benchmark dataset. The algorithm was accurate, peaking at 99.51% for TON_IoT and 98.85% for UNSW-NB15. The results show that the proposed algorithm outperformed its competitors.

IDS with an evolutionary feature selector and a RF classifier. A unique fitness function helps evolutionary techniques find relevant features and reduce data dimensionality. The real positive rate rises, whereas the false positive rate falls. Researchers employ the random forest technique for anomaly detection due to its multi-classification accuracy and noise tolerance in huge data sets. The suggested technique selects more trustworthy properties than current technologies, enhancing categorization. We provide the statistical results and method comparisons. The UNSW-NB15 and NSL-KDD datasets assess framework efficacy (Liu & Shi, 2022).

The authors in Wei et al. (2022) proposed Control Semantics-based Detection Approach (CONSTDET) which employs control semantics to detect UAV GPS spoofing attempts using ML. The research collected flight data from real trials to build CONSTDET, a viable detection system. Training GPS spoofing detection models requires carefully selecting flight data with control semantics. The data covered dynamic flight and control operations, such as altitude and horizontal position control. The authors put CONSTDET on UAV to detect GPS spoofing. The authors also trained and produced the most accurate classifier using ML. Gasimova, Khoei & Kaabouch (2022) compare bagging, stacking, and boosting ensemble-based ML algorithms. The GPS security breach tracking capabilities were assessed by prediction time per sample, memory size, processing time, PMD, PFA, and accuracy. The stacking model outperformed the other two models in accuracy, PD, PMD, and FA. The Study in Dang et al. (2022) explores deep ensemble learning and also proposes a method that detects GPS spoofing in cellular-connected UAVs. GPS-spoofing-related UAV trajectory abnormalities are identified by measuring route losses between base stations (BSs) and UAVs. To mitigate environmental impacts on path losses and ensure reliable identification, The authors use three statistical methods. MLP neural networks decide path loss statistics on the edge cloud servers. Experimental findings show that the proposed method can detect GPS spoofing with 97% accuracy with two BS and 83% with one. The proposed solution is unique because it requires no UAV energy or equipment Dang et al. (2022). The authors in Basan (2022) recommend evaluating UAV cyber-physical characteristics to identify attacks and results. A new method for producing false-attack databases and assessing their correctness is also provided in their study. To solve GPS spoofing attempts, the authors in Panice et al. (2017) used state estimation and a one-class SVM. For product efficacy and efficiency testing, they also constructed a simulation environment.

The study in Qiao, Zhang & Du (2017) suggests detecting GPS spoofing with a DJI Phantom 4 UAV’s monocular camera and IMU sensor. The authors show that incorporating these sensors can identify GPS spoofing. By analyzing the UAV’s speed using its sensors, monocular camera, and IMU, the authors can detect spoofing attacks in 5 s. The authors in Al-Wesabi et al. (2022) developed Opposition Poor and Rich Optimization-based Feature Selection with Optimal Deep Feed-forward Neural Network (OPRFS-ODFNN) to choose optimal deep feed-forward networks over poor and rich optimization-based feature selection. A new method for detecting network intrusions in IoT device communication has been developed by Al-Wesabi et al. (2022). This technique prioritizes IoD communication network safety. By scaling preliminary features and identifying important characteristics with OPRFS, OPRFS-ODFNN achieves this goal. The Optimal Deep Feed-forward Neural Network (ODFNN) model detects and classifies intrusions using Improved Mayfly Optimization (IMFO). The researchers thoroughly simulated the OPRFS-ODFNN technique to establish its usefulness. Khan et al. (2021) aims to boost UAV skills by decentralizing ML architectures with blockchain. This will enhance data integrity and storage, enabling complex, system-wide UAV decision-making. Blockchain will lead to decentralized predictive analytics and faster ML model sharing will result from blockchain. Their proposed system will demonstrate the feasibility and efficacy of collaborative intrusion detection for UAV and related applications (Khan et al., 2021).

A perfect partnership between vehicular ad-hoc networks (VANETs) and UAVs could boost network connectivity and prevent towering infrastructures, ensuring speedy data transmission. Secure VANETs and UAVs are essential. Deep learning defends VANET and UAV communications from cyberattacks that compromise data integrity, confidentiality, and availability. Bangui & Buhnova (2021) examines VANET and UAV intrusion detection systems using ML. It also addresses research gaps and suggests Intelligent Transportation System (ITS) security improvements. The study in Ahmetoglu & Das (2022) examines automatic cyber-attack identification and rapid prediction and analysis through ML. Research focuses on network traffic anomaly detection, categorization, grouping, and analysis. Each study assesses datasets, attack detection ML methods, feature selection, and dimension reduction. Ahmetoglu & Das (2022) also compares classification methods in several studies to alternatives, performance metrics, and outcomes. It analyzes open-access network attack statistics and provides a simple classification algorithm. Finally, it addresses ML’s network attack issues and offers solutions.

Yakkati et al. (2022) classifies GNSS signals using a multi-correlation receiver, examining interference-free, multi-path, jamming, and spoofing. The authors assess the test’s accuracy and confusion matrix using neural networks, SVM, KNN, kernel approximation, decision trees, discriminant analysis, naive Bayes, and ensemble classifiers. The study uses multi-correlation output to calculate average power and distortion correlation for GNSS signal categorization. Da Silva, Ferrão & Branco (2022) UAV swarms use IDS and it covers categorization, observed infiltration, ML, UAV swarm applicability, and development standards. After examining 56 relevant studies, they compared them. This study recommended IDS development. This study is significant because it was the first Database to meticulously map this topic, this study is significant.

The research in Siemuri et al. (2022) examined 2000–2021 GNSS ML studies. The authors evaluated the literature methods’ efficacy, advantages, and downsides. The investigation discovered 213 machine-learning studies. Notably, these results reveal good GNSS performance across applications. In GNSS usage scenarios, ML produces results like classical models. While ML models in GNSS have significant potential, their adoption is minimal. Promoting ML models in positioning, navigation, and timing (PNT) requires extra effort. Results reveal that hyperparameters, regularization parameters, imbalanced datasets, correlated features, and dataset size hinder ML. Using linked attributes and optimizing parameters in a balanced dataset, the Classification and Regression Decision Tree classifier obtained 99.99% accuracy. The accuracy was 99.98%, the error was 0.2%, and the false positive was 1.005%. The Random forest model has 99.94% accuracy, 99.6% likelihood of recognizing the intended outcome, 0.4% possibility of failing to detect it, and 1.01% probability of mistakenly identifying it under similar conditions. The study in Dhal & Azad (2022) investigates FS extensively. It details FS frameworks and models. FS algorithms process structured, labeled, or unstructured data. The study covers FS principles, common approaches, popular datasets, and important contributions from several ML domains relevant to FS applications. This descriptive survey examines FS’s concepts and application in different issue categories. Overall, the presentation is good in Dhal & Azad (2022).

Aissou et al. (2021) suggest comparing tree-based ML methods such as RF, Gradient Boost, XGBoost, and LightGBM to detect GPS spoofing attacks to address the security issue. XGBoost surpassed the others with 95.52% accuracy and 2 ms detection. These findings imply that XGBoost’s precision and speed could help UAS. The article Talaei Khoei, Ismail & Kaabouch (2022) proposes a one-stage feature selection technique to remove associated and low-importance dataset features. To determine the optimum assault classifier, this study introduced metric-optimized and weighted dynamic selectors to determine the optimum assault classifier. We assess ten ML models for accuracy, detection, false detection, and processing time. Our innovative method dynamically chooses the best classifier, surpassing ensemble models. This yields 99.6% accuracy, 98.9% detection likelihood, 1.56% FA likelihood, 1.09% misdetection likelihood, and 1.24 s of processing time.

This article offers significant improvements that surpass current research on GPS spoofing attack detection. It has a hybrid model that combines bidirectional long short-term memory (BiLSTM) networks with convolutional neural networks (CNN). This makes detection more accurate than with some machine learning methods. The research utilizes extensive feature selection methods, such as ANOVA, to identify the most important features for detecting GPS spoofing, thereby overcoming the shortcomings of inadequate feature selection in previous studies. We employ stringent oversampling approaches to address the prevalent issue of imbalanced datasets and ensure reliable detection performance. The suggested hybrid model exhibits exceptional performance measures, attaining a precision of 98.84%, an accuracy of 99.25%, an F1 Score of 99.26%, and a recall of 99.69%, signifying significant advancements over current methodologies. The dataset includes different types of GPS spoofing attacks, such as simple spoofing, in which fake GPS signals confuse the UAV’s navigation system, intermediate spoofing, in which GPS signals are changed gradually to control the UAV, and advanced spoofing, in which the attacker can fake real signals to get around normal detection methods. The proposed architecture proposes extensive detection capabilities by tackling these attack types, rendering it resilient to various GPS spoofing scenarios.

In recent years, the safety of multi-UAV systems (MUSs) has garnered heightened attention due to increasing threats from composite attacks, including denial-of-service (DoS) attacks, false data injection (FDI) attacks, camouflage attacks, and actuation attacks (AAs). In order to mitigate these advanced threats, Gong et al. (2023) introduced a robust two-tiered system. The framework uses a digital twin architecture to divide the defenses into two levels: a twin layer (TL) to protect against DoS attacks and a cyber-physical layer (CPL) to handle unbounded AAs. When a topology-repairing method is combined with a decentralized adaptive controller, uniformly ultimately bound (UUB) convergence is guaranteed. This makes multi-agent systems (MUSs) more resistant to composite attacks. Protecting multi-agent systems (MASs) from malicious threats, including Byzantine assaults, is a persistent concern in distributed systems. Gong et al. (2024b) created a Distributed Byzantine-Resilient Observer (DBRO) that makes sure that high-order multi-agent systems (MASs) working on directed graphs converge to zero errors in finite time. This edge-based DBRO system assesses the leader’s state inside a multi-agent network while mitigating Byzantine threats. Robust graph theory underpins the methodology’s cascading architecture, which enhances its applicability to both time-invariant and time-varying topologies, thereby demonstrating notable resilience against Byzantine agents (Gong, Li & Shu, 2024). The emphasis on multi-agent systems (MASs) and UAV swarms has brought considerable attention to security issues, especially for human-in-the-loop (HiTL) systems susceptible to Byzantine assaults. Gong et al. (2024a) created a strong two-tier hierarchical control system that protects against Byzantine edge attacks (BEAs) and Byzantine node attacks (BNAs). It uses a digital twin layer (DTL) and a cyber-physical layer (CPL). The architecture enables a human operator to direct a non-autonomous leader UAV while mitigating these hostile assaults. The results of their experiments showed that the control framework worked well at protecting multiple UAV systems. This suggests that resilient swarm robotics could be useful in dangerous situations (Gong et al., 2024a). Multi-agent systems (MAS) have made significant progress in resilient control, particularly in preventing Byzantine attacks, which pose a significant threat due to their ability to disseminate false information through compromised agents. Gong et al. (2024b) presented an innovative twin-layer hierarchical control scheme that separates the defense mechanisms for Byzantine Node Attacks (BNAs) and Byzantine Edge Attacks (BEAs). Their method uses a DTL to fight BEAs and a CPL with a decentralized adaptive controller to fight BNAs. This unique design improves system resilience by implementing a trusted-node method to protect important nodes and applying a chattering-free control scheme for efficient response to diverse attack scenarios. Compared to other protection methods, this two-layer strategy guarantees better performance for both compromised and benign agents. It sets a new standard for multi-agent system security in hostile environments. Path planning for UAVs in hostile situations is essential, especially when confronting clandestine assaults that alter sensor data or control inputs undetected. He et al. (2023) present a strong path planning method to protect ultrawideband (UWB) sensors from hidden attacks. They use Pontryagin’s maximal principle and think of the problem as a Stackelberg game. This method enables the UAV to calculate energy-efficient routes while predicting and alleviating the impacts of covert assaults. Their approach is innovative in tackling vulnerabilities in UWB sensors and enhancing resilience through game theory, beyond conventional tactics that concentrate on detectable threats such as GPS spoofing. This study makes UAVs safer by simulating how an attacker and a defender interact with each other. It then offers ways to defend against sneaky attacks while still completing missions in both single base station (SBS) and double base station (DBS) settings. These advances have tangible implications for improving UAV resilience and operational integrity, particularly in military, surveillance, and emergency response contexts.

By conducting in-depth research and thorough analysis, it has become evident that the detection of GPS spoofing attacks is of paramount importance due to their potential to compromise critical systems reliant on accurate positioning information. The need for a new framework arises from the limitations of existing methods in effectively detecting GPS spoofing attacks. Current approaches often struggle with minimizing detection time, ensuring transferability, and reducing false negatives. Moreover, they may lack the capability to handle complex attacks that manipulate ambient parameters or duplicate natural trajectories without sufficient context. These limitations highlight the necessity for a more comprehensive and efficient framework, which prioritizes dataset balancing, feature extraction, noise removal, and NN efficiency to address these challenges effectively.

Proposed model

By conducting in-depth research and thorough analysis, a new conceptual framework called DeepSpoofNet is being developed, as illustrated in Figs. 2 and 3. This framework addresses the challenges of detecting GPS spoofing attacks by focusing primarily on minimizing detection time while simultaneously enhancing transferability and reducing false negatives. To detect GPS spoofing attempts, the DeepSpoofNet framework increases dataset balancing, feature extraction, noise removal, and NN efficiency. The suggested UAV GPS spoofing attack detection paradigm cleans and sanitizes flight log data. Initially, null values and duplicate data are removed to build a reliable dataset. The sanitized data is balanced in the second phase to address class imbalances and assure target class representation. This stage helps the model understand minority and majority-class events. Earlier research has detected GPS spoofing attacks using local variables x, y, z, vx, vy, and vz. However, complex attacks that alter ambient parameters or duplicate natural trajectories may be harder to discern without context, although they capture crucial positional dynamics. To overcome this constraint and strengthen our technique, we include both local and global variables, such as latitude, longitude, altitude, and sensor data.

Figure 2 DeepSpoofNet’s proposed model diagram.

Figure 3 DeepSpoofNet’s proposed model abstract diagram.

A larger dataset provides a more comprehensive assessment framework. Global factors and sensor data allow the algorithm to distinguish manipulated signals from normal signals, enhancing GPS signal behavior knowledge. We prioritize processing cost reduction and detection accuracy. Sensor data, as well as local and global variables, help the system detect subtle and complex GPS spoofing attacks. The third phase, feature analysis, selects features optimally based on the data balance. The suggested framework explores the best combination of FS and DL approaches to tackle the crucial problem of identifying GPS spoofing attacks on unmanned aerial vehicles. By altering GPS signals, GPS spoofing can result in mission failures or unexpected behavior, which is a serious danger to UAV operations. By analyzing how FS techniques and DL algorithms interact and determining the best combinations for performance, this work improves detection systems. A significant area of attention is the problem of unbalanced datasets, which is a frequent obstacle in GPS spoofing detection. When spoofing attempts are neglected, biased models resulting from imbalanced data frequently fail to identify them. The system uses sophisticated oversampling techniques to balance the dataset in order to address this. In order to preserve just the most essential features for spoofing detection, the framework also gives priority to efficient FS. This enhances the precision and resilience of DL models while lowering computational complexity. The framework advances security frameworks and ensures robust operation in hostile contexts by successfully merging FS and DL to provide a scalable and dependable solution for GPS spoofing detection on UAVs.

The DeepSpoofNet method finds the best model training features, enhances accuracy, and minimizes the true negative rate. By selecting the most discriminative attributes, the model focuses on UAV GPS spoofing attack detection. After curating and optimizing the dataset, we train a NN model. To detect UAV GPS spoofing, the NN learns complex patterns and correlations from features to identify UAV GPS spoofing. A trained model detects and mitigates GPS spoofing risks for real-time UAV security. The model detects UAV GPS spoofing attacks utilizing data cleaning, balancing, feature analysis, NN training, and deployment. Data quality, class imbalance, feature relevance, and NN allow the model to detect GPS spoofing attacks and secure UAV systems. To the best of our knowledge, we provide the best techniques and strategies for each model phase. Accuracy, computational needs, and robustness determine the optimum data cleaning, data balancing, feature analysis, and NN training techniques. This research improves UAV GPS spoofing attack detection, allowing for more robust and efficient security and operational integrity detection systems.

As shown in Fig. 4, this model uses convolutional, pooling, normalizing, and recurrent layers to detect GPS spoofing in UAVs. The first step involves transforming the data to align with the input format of the convolutional layer. The first Conv1D layer uses 64 filters to detect local GPS signal characteristics, such as frequency or intensity shifts, that could indicate spoofing. With MaxPooling, you can reduce dimensionality to focus on key features and boost computing performance. Handling raw GPS data with large-scale or noise differences requires batch normalization to stabilize the learning process by normalizing the output.

Figure 4 DeepSpoofNet’s proposed model detail diagram.

A second Conv1D layer with 128 filters extracts more complicated signal properties, and max pooling reduces data size while maintaining crucial information. The Bidirectional LSTM layer and other recurrent neural networks look at time relationships from both directions. This is crucial for identifying spoofing attacks, which modify timing and signals. Flattening multi-dimensional data into a 1D vector prepares it for fully connected layers. The Dense layer processes these features, producing the sigmoid output layer that detects GPS signal spoofing. To capture GPS data’s local and global patterns, the architecture uses convolutional layers for feature extraction and LSTMs for time-series data. This method helps the model detect spoofing attempts in complex or imbalanced datasets.

Results and Discussions

The experimental inquiry includes preparing and analyzing UAV flight log data, creating and setting up the model architecture, and testing the model’s performance. The datasets from genuine UAV flights cover a wide range of authentic and practical settings, ensuring the model’s robustness and usefulness in real-world situations.

Hardware/software used

In this study, the experimental PC utilized is equipped with an Intel Core 2.4 GHz i5 6th generation processor, 8 GB of RAM, and a 256 GB hard disk. Prioritizing resource efficiency and accessibility, the experiment was conducted without a graphics card. Anaconda Navigator serves as the management tool for packages and environments in our lab, eliminating the need to manually enter conda commands in a terminal. For practical studies, Jupyter Notebook is employed, offering a user-friendly interface for code development and analysis. Python serves as the primary programming language, ensuring compatibility and versatility for research experiments. This integrated approach enhances workflow, facilitating efficient experimentation and code creation within the research environment. A diverse range of libraries are utilized in the practical study to enable effective data manipulation, analysis, and ML. Numpy 1.26.0 (van der Walt, Colbert & Varoquaux, 2011) serves as a foundational tool for numerical operations and array manipulations, along with essential scientific computing functionalities. Pandas 2.1.1 (McKinney, 2010) simplifies structured data management and preparation for exploratory data analysis. Seaborn 0.13.0 (Waskom, 2021) and Matplotlib 3.8.0 (Hunter, 2007) are employed for creating visually appealing plots and graphs that elucidate experimental results. TensorFlow 2.14.0 (Abadi, 2016) and Keras 2.14.0 (Chollet, 2015) support the implementation of complex neural network models, while scikit-learn 1.3.1 (Pedregosa, 2011) provides reliable ML algorithms and model assessment tools. The experimental framework is constructed around this carefully selected library collection to ensure the attainment of dependable and efficient study goals.

Dataset description

The study’s dataset comprises a carefully curated collection of both synthetic and real-world data, capturing a diverse array of scenarios essential to the study’s objectives. This comprehensive dataset ensures a thorough examination of the proposed hypotheses. To maintain data integrity and enable trustworthy experimentation and analysis, rigorous preparation techniques were meticulously applied. This meticulous preparation not only safeguards data integrity but also enhances the reliability of the experimental process. The UAV Attack dataset (Whelan et al., 2020), provided by the IEEE for research in UAV security, aggregates instances of UAV attacks orchestrated by various jammers. Table 5 outlines the features used in this study, while Table 6 illustrates the ratio of benign to attacked data.

Table 5 Description of features.

#	Feature	Description	
1	timestamp	Time at which the data is recorded	
2	global_lat	Latitude coordinate of the GPS position	
3	global_lon	Longitude coordinate of the GPS position	
4	global_alt	Altitude above sea level of the GPS position	
5	global_eph	Estimated horizontal position error of the GPS data	
6	global_epv	Estimated vertical position error of the GPS data	
7	lo_x	X-coordinate of local position	
8	lo_y	Y-coordinate of local position	
9	lo_z	Z-coordinate of local position	
10	lo_vx	Velocity component in the x-direction of local position	
11	lo_vy	Velocity component in the y-direction of local position	
12	lo_vz	Velocity component in the z-direction of local position	
13	lo_ax	Acceleration component in the x-direction of local position	
14	lo_ay	Acceleration component in the y-direction of local position	
15	lo_az	Acceleration component in the z-direction of local position	
16	lo_eph	Estimated horizontal position error of local position data	
17	lo_epv	Estimated vertical position error of local position data	
18	lo_evh	Estimated horizontal velocity error of local position data	
19	lo_evv	Estimated vertical velocity error of local position data	
20	alt_ellipsoid	Altitude above ellipsoid	
21	s_variance_m_s	Speed variance	
22	c_variance_rad	Course variance	
23	hdop	Horizontal dilution of precision	
24	vdop	Vertical dilution of precision	
25	noise_per_ms	Noise per millisecond	
26	jamming_indicator	Indicator of jamming	
27	vel_m_s	Velocity magnitude	
28	vel_n_m_s	Velocity in the north direction	
29	vel_e_m_s	Velocity in the east direction	
30	vel_d_m_s	Velocity in the down direction	
31	cog_rad	Course over ground in radians	
32	timestamp_time_relative	Time relative to the start of the dataset	
33	heading	Aircraft heading	
34	heading_offset	Offset of heading	
35	vel_ned_valid	Validity of velocity in North, East, Down directions	
36	satellites_used	Number of satellites used in the fix	
37	mag[0]	Magnetic field strength in the x-direction	
38	mag[1]	Magnetic field strength in the y-direction	
39	mag[2]	Magnetic field strength in the z-direction	
40	baro_alt_meter	Barometric altitude	
41	baro_temp_celsius	Barometer temperature in Celsius	
42	baro_pressure_pa	Barometric pressure in Pascals	
43	rho	Air density	
44	gyro_rad[0]	Gyroscopic angular velocity around the x-axis	
45	gyro_rad[1]	Gyroscopic angular velocity around the y-axis	
46	gyro_rad[2]	Gyroscopic angular velocity around the z-axis	
47	acc_m_s2[0]	Acceleration in the x-direction measured by accelerometers	
48	acc_m_s2[1]	Acceleration in the y-direction measured by accelerometers	
49	acc_m_s2[2]	Acceleration in the z-direction measured by accelerometers	

Table 6 Dataset before and after balancing.

Comparison	The ratio of benign to attacked data	
Before oversampling	1:4	
After oversampling	1:1	

Data pre-processing

This research employed various essential strategies during the preprocessing phase to achieve data balancing, cleaning, and feature selection. Initially, we utilized SMOTE (Synthetic Minority Oversampling Technique) to rectify class imbalance within the dataset. Given the infrequency of GPS spoofing attempts relative to standard data points, SMOTE facilitated the creation of synthetic samples for the minority class. This ensured the equilibration of the dataset and provided the model with sufficient examples of both authentic and counterfeit data, thereby boosting its ability to generalize across categories. After balancing the data, we sanitized the dataset to remove any null or missing values, which ensured uniformity across all characteristics. This stage was crucial for preventing inadequate information during the model training process, which could result in inaccurate predictions or biased outcomes. We employed various statistical methods for feature selection to reduce complexity and enhance the effectiveness of the model. We employed PCA to transform the dataset into a lower-dimensional space, thereby maximizing variance and reducing redundancy. Furthermore, we assessed feature relevance using chi-square tests, Pearson correlation coefficient (PCC), and ANOVA (Analysis of Variance), each providing distinct insights into the relationship between characteristics and the objective variable. These strategies facilitated the elimination of irrelevant or redundant features, preserving the most informative information for model training. We ultimately applied the model to the altered datasets produced by each feature selection method (PCA, chi-square, PCC, and ANOVA). We subsequently compared the results using the model’s performance measures, which included accuracy, precision, recall, and F1-score. We chose the optimal combination of features and preprocessing techniques for subsequent model optimization, ensuring the use of the most efficient configuration in the final implementation.

Feature selection

Developing a reliable and effective model to identify GPS spoofing attempts on UAV requires feature selection (FS). The study uses PCA, chi-square, PCC, and ANOVA for FS. PCA, a popular linear dimension reduction method, was used to find the dataset’s most variable components. To choose the most important features for classification, the chi-square statistical test was used to evaluate the features and the target variable. The linear link between each attribute and the target variable was assessed using the PCC. A stronger correlation indicates greater relevance. ANOVA was performed to determine how much each component explained target variable variability, with smaller p-values indicating greater relevance. Our FS method reduces the dataset’s dimensionality and identifies the best characteristics for building a robust and accurate GPS spoofing detection model for UAV. The recommended detection model is trained and tested using the selected attributes, outperforming current methods, and shows how FS improves UAV GPS-spoofing attack detection systems.

Model evaluation

The framework requires model building following dataset preparation. Predict multiple modalities for effective classification. The article uses CNN and ANN deep learning algorithms. These models are used with chi-square, PCC, ANOVA, and PCA to find the best UAV GPS spoofing assault detection approach. Industry guidelines require us to split the dataset 70% for model training and 30% for testing. Because they work well in binary datasets, this study’s hyper-parameter settings use sigmoid and RELU activation functions. We optimize the models with ADAM because of its strength and noise tolerance. The loss function is binary cross-entropy since GPS spoofing detection is binary. We alter hyper-parameters, especially epoch fluctuation, to discover the optimal GPS spoofing attack detection epoch configuration. Activation functions and optimizers are distinguished by their binary dataset and GPS spoofing detection capabilities. This technique optimizes deep learning model hyper-parameters and FS. Dataset experiments test the framework’s GPS spoofing detection. Our machine learning model evaluation criteria prompted us to divide the dataset into 70% training and 30% testing sets. This section employs two deep-learning models and four feature extraction methods. This study evaluates GPS signal alteration detection models. The dataset’s training helps algorithms identify benign and spoofed patterns. Chi-square, PCC, ANOVA, and PCA extract characteristics from two deep learning models with distinct architectures. A detailed analysis of the models’ performance across feature sets shows their strengths and flaws. To discover the optimal GPS spoofing detection setup, this study changes feature extraction models and approaches.

Comparative analysis

This comprehensive study compares GPS spoofing attack detection studies using classic ML and advanced deep learning models. The early attempts used ANN models with varied FS strategies.

ANN with FS methods

ANN performance with PCA, PCC, Chi-Squared test, and ANOVA helps us detect GPS spoofing assaults. PCA gives the model 97.48% accuracy, exhibiting good performance. The PCA is less precise than other methods. PCA captures variance effectively; however, it may not be as excellent at identifying subtle GPS spoofing patterns. Table 7 shows ANN with different FS methods. GPS spoofing detection concerns arise from the PCC’s 80.91% precision. Lower precision increases the likelihood of declassifying irrelevant cases, indicating feature interaction issues. A 91.12% precision and 94.72% accuracy make the chi-squared test good.

Table 7 Performance metrics: ANN with different FS methods.

FS Method	Precision	Accuracy	F1 score	Recall	
PCA	97.07	97.48	97.52	97.97	
PCC	80.91	86.34	87.59	95.48	
Chi-squared test	91.12	94.72	95.00	99.22	
ANOVA	95.99	97.40	97.47	98.99	

ANN with FS method maintains accuracy while identifying patterns, striking a balance between precision and recall as shown in Table 7, and Fig. 5. The chi-squared test produces competitive GPS spoofing detection (FS) results. ANOVA consistently performs well, exhibiting 95.99% precision and 97.40% accuracy. ANN with FS technique aids in detecting GPS spoofing by uncovering subtle group deviations. A thorough analysis underscores the importance of evaluating FS methods in conjunction with ANN. While PCA is accurate, its precision may limit practicality. On the other hand, the PCC, offering lower precision, may not be suitable for applications requiring high accuracy in GPS spoofing detection. The balanced performance of the chi-squared test is intriguing for precision and memory considerations. With its precision and accuracy, ANOVA emerges as the optimal FS method for robust GPS spoofing detection within the ANN framework. Particularly adept at identifying GPS spoofing trends, ANOVA can unveil significant changes and connections across variables.

Figure 5 Comparison of metrics graphs for ANN.

CNN with FS methods

This CNN-based FS study has enhanced GPS spoofing attack detection. The comparison underscores crucial performance parameters, such as model efficacy and accuracy as shown in Table 8, and Fig. 6. Notably, CNN and PCA demonstrated a tradeoff with 20 and 18 features, respectively. Utilizing 18 features, the model exhibited adaptability to a smaller feature space, achieving 96.38% precision and 97.87% accuracy. This adaptability ensures efficient calculation and resource utilization. However, the PCC significantly diminished CNN performance measures, showcasing challenges in understanding complex feature interactions. Linear correlation alone may prove insufficient for identifying GPS spoofing. Conversely, the CNN with chi-squared test effectively identifies variable dependencies, yielding 99.34% precision and 96.89% accuracy. Nonetheless, the lower F1 score and recall may pose challenges in balancing all metrics, potentially obscuring critical data trends. On the other hand, CNN ANOVA FS consistently delivered strong performance, with an F1 score, precision, accuracy, and recall of 98.23%, 98.94%, 98.96%, and 99.69%, respectively.

Table 8 Comparative analysis of CNN with PCA, PCC X2 and ANOVA.

FS method	Precision	Accuracy	F1 Score	Recall	
PCA	96.38	97.87	97.93	99.53	
PCC	77.49	84.29	86.20	97.12	
Chi-squared test	99.34	96.89	96.84	94.47	
ANOVA	98.23	98.94	98.96	99.69	

Figure 6 Comparison of metrics graphs for CNN.

This group variance-focused method gathered complex GPS signal data patterns with several features. The comparison study concludes that CNN FS is crucial for GPS spoofing detection. Balance is needed because the chi-squared test was precise but difficult to recall. With its vast feature set and excellent performance across all parameters, ANOVA was the most robust method. This work informs the design and optimization of GPS spoofing detection systems.

CNN-BiLSTM with FS methods

GPS spoofing attack detection using CNN-BiLSTM with various FS methods results in diverse performance metrics, as depicted in Table 9 and Fig. 7. Leveraging PCA’s 18 attributes, the model exhibits strong performance with 97.17% precision, 98.11% accuracy, 98.15% F1 score, and 99.14% recall. This underscores the effectiveness of PCA in gathering data for accurate categorization. Conversely, CNN-BiLSTM demonstrates subpar performance with PCC FS. The model achieves a precision of 78.77%, an accuracy of 86.30%, an F1 score of 88.03%, and a recall of 99.77% for GPS spoofing attacks, contrasting with PCA. Here, the linear relationship of PCC may not offer meaningful insights. Conversely, employing the Chi-Squared test for feature selection enhances precision, accuracy, F1 score, and recall by 93.17%, 96.30%, 96.47%, and 100.00%, respectively. This underscores the significance of feature independence in the model’s ability to detect subtle GPS modification patterns. Employing ANOVA characteristics, CNN-BiLSTM achieves 99.25% accuracy, 98.84% precision, 99.26% recall, and a 99.26% F1 score, leading to improved performance. This highlights the capability of ANOVA in identifying classification-relevant traits. The comparison underscores the impact of FS on CNN-BiLSTM GPS spoofing attack detection performance.

Table 9 Comparative analysis of CNN-BiLSTM with different FS methods.

FS method	Precision	Accuracy	F1 score	Recall	
PCA (18 Features)	97.17	98.11	98.15	99.14	
PCC	78.77	86.30	88.03	99.77	
Chi-squared test	93.17	96.30	96.47	100.00	
ANOVA	98.84	99.25	99.26	99.69	

Figure 7 Comparison of graphs for CNN-BiLSTM.

Table 9 shows results, while Fig. 7 show graphs. ANOVA emphasizes essential feature contributions; chi-squared tests feature independence; and PCA captures overall variance. These results drive FS and implementation to improve CNN-BiLSTM GPS spoofing detection accuracy.

Overall best-performed models with FS methods

We tested different FS methods and deep learning model combinations to find the best GPS spoofing attack detection method. ANN and CNN were integrated with PCA and ANOVA for FS. The experiment produced ANN outcomes with 97.07% precision, 96.48% accuracy, 97.52% F1 score, and 97.97% recall using 20 PCA-selected features. This method provided a baseline for comparison and demonstrated PCA’s ability to extract ANN features. CNN and ANOVA indicated considerable improvements: precision 98.23%, accuracy 98.94%, F1 score 98.96%, and recall 99.69%. The CNN architecture detected GPS signal to fake better with ANOVA’s discriminative properties. Out of all the methods tested, CNN-BiLSTM with ANOVA worked best. The hybrid model had 98.84% precision, 99.25% accuracy, a 99.26% F1 score, and a 99.69% recall. GPS spoofing detection was more predictive using ANOVA-selected features and CNN and BiLSTM layers. The comparison of these results underscores the importance of hybrid models, which combine deep learning architectures with well-chosen features. ANN and PCA work well, but CNN and BiLSTM layers with ANOVA-selected features provide unprecedented detection accuracy.

This investigation illustrates the optimal combinations and underscores the importance of tailoring model architectures to feature attributes for enhancing GPS spoofing detection systems. Tables 10 and 11 respectively present a comparative analysis of the best performing GPS spoofing detection models and complexity analysis of deep learning models with PCA, PCC, X2, and ANOVA. Additionally, Fig. 8 depict the results. We used a methodical approach to choose models by combining PCA, chi-square, PCC, and ANOVA feature selection methods with CNN-BiLSTM and CNN models. Both models assessed every dataset with selected features. The CNN-BiLSTM integrated CNN for feature extraction with BiLSTM to identify sequential patterns, whereas the independent CNN concentrated on spatial feature extraction. We evaluated performance using accuracy, precision, recall, and F1 score for each combination. These metrics guided the selection of the most effective feature selection approach and model architecture, ensuring maximum accuracy and resilience for the GPS spoofing detection framework.

Table 10 Comparative analysis of best performed GPS spoofing detection models.

Model	Precision	Accuracy	F1 score	Recall	
ANN with PCA	97.07	97.48	97.52	97.97	
CNN with ANOVA	98.23	98.94	98.96	99.69	
CNN-BiLSTM with ANOVA	98.84	99.25	99.26	99.69	

Table 11 Complexity analysis: deep learning models with PCA, PCC, X2 and ANOVA.

Algorithm	FS method	Training time ( μs)	
ANN	PCA	9,191.75	
ANN	PCC	10,539.77	
ANN	X2	7,184.03	
ANN	ANOVA	7,838.01	
CNN	PCA	8,655.07	
CNN	PCC	10,158.06	
CNN	X2	9,617.09	
CNN	ANOVA	14,127.25	
CNNBi-LSTM	PCA	10,164.02	
CNNBi-LSTM	PCC	10,396.96	
CNNBi-LSTM	X2	8,592.84	
CNNBi-LSTM	ANOVA	7,005.93	

Figure 8 Comparison of graphs for best performed GPS spoofing detection models and complexity analysis.

Comparative analysis with state-of-the-art research

UAVs are difficult to detect GPS spoofing, although machine learning models like DeepSpoofNet, Wu et al. (2023), and Wei et al. (2022) offer several solutions. DeepSpoofNet, our model, performs better due to its complex design, extensive feature selection, and effective data imbalance control. CNNs and BiLSTM layers make up DeepSpoofNet’s hybrid design. In this integration, the model can detect geographical and temporal correlations in GPS signal data, effectively distinguishing legitimate signals from fakes. The CNN-BiLSTM design catches detailed GPS spoofing signal distortions, making it more robust to many assault situations. DeepSpoofNet selects only the most important and relevant features using ANOVA. Reducing noise and focusing the model on key data improves forecast accuracy. With the synthetic minority oversampling technique, DeepSpoofNet reduces class imbalances. This helps the model learn from dominant and minority classes—even rare spoofing events. DeepSpoofNet performs well by training on a large dataset that comprises self-experiment data and IEEE Dataport data. It has a spectacular 99.25% detection accuracy, 98.84% precision, 99.69% recall, and 99.26% F1 score. These results show that DeepSpoofNet is the best GPS spoofing detection technology. Wu et al. (2023) built a CNN-BiLSTM model to emphasize GPS signal spatial and temporal characteristics. Table 12 provides a detailed comparison of the state-of-the-art methods.

Table 12 Comparative analysis with state-of-the-art research.

Study	FS	Acc.	Prec.	Rec.	F1	Data	
DeepSpoofNet	ANOVA	99.25	98.84	99.69	99.26	Whelan et al. (2020)	
Wu et al. (2023)	SHAP	99.1	N/A	N/A	N/A	Sim.	
Wei et al. (2022)	DT	97.70	98.70	96.76	97.72	Realtime dataset	

In contrast, Wei et al. (2022) uses XGBoost, a gradient-boosting algorithm that works well with structured data. For GPS spoofing detection, Wei et al. (2022) prioritize flight dynamics—latitude, longitude, altitude, velocity, and orientation angles (roll, pitch, yaw). Wei et al. (2022) model modifies the flight dataset for UAV activities, making it domain-specific. Although XGBoost manages structured data well, Wei et al. (2022) model is less accurate than DeepSpoofNet. Because of its hybrid design, feature selection, and class imbalance mitigation, DeepSpoofNet outperforms Wu et al. (2023) and Wei et al. (2022) DeepSpoofNet uses CNN-BiLSTM layers, ANOVA feature selection, and SMOTE to provide the most accurate and reliable GPS spoofing available. It is the best way to protect UAVs against GPS spoofing.

Conclusion and future work

This study has investigated various ML, deep learning, and non-ML methods for detecting UAV GPS spoofing. Previous models were found to have several limitations, which were thoroughly examined in this research. While ML algorithms for detecting UAV GPS spoofing attacks have been extensively researched, challenges such as unbalanced datasets and inadequate FS for model training persisted in experimental approaches. Through a deep technical comparison, this article has identified the most effective model training algorithms for GPS spoofing detection. The proposed DeepSpoofNet framework, utilizing NN and filtering, represents the convergence of technology, security, and artistic expression in GPS spoofing attack detection. By employing balanced datasets to reduce bias and enhance model training, filtering to eliminate noise and determine UAV state, and FS and dataset extraction to improve model performance and efficiency, this framework addresses the limitations of current GPS spoofing attack detection methods on UAVs. Specifically, it tackles strategies for managing imbalanced datasets, optimal feature engineering, and key performance analysis parameters for model effectiveness. Upon reviewing the most effective GPS spoofing attack detection methods for UAVs, it became evident that the limited range of features was a significant constraint. Local features such as position and velocity were prioritized, hindering proper identification. To overcome this limitation, a novel model was introduced that utilizes a wider range of GPS and other sensor properties, resulting in improved accuracy and robustness in GPS spoofing attack detection. The examination encompassed the utilization of a GPS spoofing attack dataset, normalization techniques using NumPy and Pandas, as well as addressing data imbalance through SMOTE. FS methods such as PCA, PCC, chi-squared test, and ANOVA were meticulously paired with three deep learning models namely ANN, CNN, and CNN-BiLSTM. Evaluation criteria included model performance metrics such as accuracy, F1-score, recall, and precision. The comparative analysis shed light on the efficacy of different model architectures and FS configurations, revealing superior performance in detecting GPS spoofing assaults on UAV with certain combinations. For instance, the CNN-BiLSTM with PCA combination utilizing 18 features demonstrated 97.17% precision, 98.11% accuracy, 98.15% F1-score, and 99.14% recall. In conclusion, this study advances the field of UAV GPS spoofing assault detection by delineating the varied efficacy and performance of FS strategies and deep learning models. This study represents a significant advancement in UAV GPS-spoofing assault detection. By addressing earlier approach flaws and incorporating innovative alternatives, the proposed model enhances accuracy and robustness. Through rigorous methodologies including SMOTE for managing unbalanced datasets and investigation of feature engineering strategies, this study provides valuable insights for improving UAV security against GPS spoofing attacks.

Future research and development in UAV GPS spoofing attack detection presents several avenues for exploration. Real-time data streaming can enhance model adaptability to dynamic environments while integrating additional environmental factors like signal intensity and weather conditions could bolster model robustness. Ensemble techniques that amalgamate the strengths of multiple models may yield more accurate and resilient detection systems. Further optimization of architectural layouts and hyperparameters tailored to specific scenarios could enhance model performance. Additionally, investigating the impact of attack complexity on model efficacy and studying adversarial assaults on GPS spoofing detection models are imperative for fortifying systems against potential threats.

Additional Information and Declarations

Competing Interests

Sedat Akleylek is an Academic Editor for PeerJ.

Author Contributions

Aziz Ur Rehman Badar conceived and designed the experiments, performed the experiments, performed the computation work, authored or reviewed drafts of the article, and approved the final draft.

Danish Mahmood performed the experiments, performed the computation work, authored or reviewed drafts of the article, and approved the final draft.

Adeel Iqbal analyzed the data, prepared figures and/or tables, and approved the final draft.

Sung Won Kim analyzed the data, authored or reviewed drafts of the article, and approved the final draft.

Sedat Akleylek performed the experiments, prepared figures and/or tables, and approved the final draft.

Korhan Cengiz conceived and designed the experiments, authored or reviewed drafts of the article, and approved the final draft.

Ali Nauman conceived and designed the experiments, analyzed the data, prepared figures and/or tables, and approved the final draft.

Data Availability

The following information was supplied regarding data availability:

The raw data is available at Zenodo: Aziz Ur Rehman Badar. (2024). DeepSpoofNet: A Framework for Securing UAVs against GPS Spoofing Attacks (1.0). Zenodo. https://doi.org/10.5281/zenodo.13963838.

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
