# Peer review of "DeepSpoofNet: a framework for securing UAVs against GPS spoofing attacks"

_PeerJ Computer Science, doi:10.7717/peerj-cs.2714_

## Round 0.1 · original submission · Major Revisions

Our apologies for a delayed decision. We have now received 3 reviews - please address their comments in a revision.

Reviewer 1 ·

Basic reporting

There were some textual editing errors in the paper.

Experimental design

Section 3 did not clearly explain the proposed method, and it was necessary to improve the method description.

This paper did not mention the source of the dataset. The dataset included various data types, among which certain data were selected as features.

This paper addressed defense against GPS spoofing attacks, but did not mention the model and configuration of the UAV used in the experiments.

It was necessary to analyze whether the proposed model could be deployed on a real UAV and evaluate its attack detection performance.

Validity of the findings

The innovation points of this paper were not clearly defined. Compared to existing work, what are the advantages of the proposed method? Can these be demonstrated through experimental comparison?

Cite this review as

Reviewer 2 ·

Basic reporting

The paper proposes a framework for securing UAVs against GPS spoofing attacks. It is readable but there are several concerns:

1) what is the contribution of this study, compared to other studies in literature. What types of GPS spoofing attacks are available on this data?


2) The data pre-processing, the model selection, needs more clear description and justification


3) The reviewer suggests to add a table at the end of results to discuss and compare the result of this proposed models with other models in literature.


4) The grammar and English need double check

Experimental design

The authors needs to describe the exact experimental design they had during the model simulations.

Validity of the findings

The authors have to provide a table and section to discuss and analyze the results of their proposed model.

Additional comments

N/A

Cite this review as

Reviewer 3 ·

Basic reporting

I have read the paper with great interest, and I found the technical results presented to be correct. However, I am not satisfied with the overall presentation, which is recommended to have a major revision. In my opinion, this manuscript can be improved via considering the following aspects.

1. In the second part of this article, the literature review, it is evident that the author has conducted extensive research. However, the text description and tables are repetitive, and the content is lengthy. It is therefore recommended that the content be streamlined and repetition avoided.
2. The model proposed in the third part should be described in detail, point by point, e.g. How is the detection time minimised? Which mathematical formulas are used? What are the mathematical derivations?
How are dataset balancing, feature extraction, noise removal, etc. added to the DeepSpoofNet framework? How is the relationship between them regulated? What is the difference with the existing results or just integration? Please explain it in detail.
3. In the dataset description section in 4.2, only the selection of the dataset is described objectively. However, in the third section it is mentioned that the DeepSpoofNet framework adds data set balancing to detect GPS attempts. How is this data set balanced? What existing results or mathematical knowledge are used and what improvements are made? Please explain it in detail.
4. In the comparative analysis of Part 5, I can only see the data comparison of different methods, which lacks real verification and whether a real simulation experiment has been conducted. It is recommended to first add the simulation experiment result graphs, such as the accurate recognition rate of each method, and then compare them separately.
5. It is recommended to modify the overall structure of the article, make it more detailed and add more descriptions of the proposed model.
6. Some of the latest techique advances concerning spoofing attacks should be discussed and compared, such as [R1, doi:10.1109/TAC.2023.3321866],[R2, doi:10.1109/TCYB.2023.3281902],[R3, doi: 10.1109/JAS.2023.123339],[R4, doi: 10.1109/TII.2023.3240595], and [R5, doi: 10.1109/TASE.2024.3400155].

7. It is suggested to have a double check on the writing details throughout the whole paper.

Experimental design

More experiment details should be added, to show the effectiveness.

Validity of the findings

Need to be improved via Literature discussions and experiment details.

Additional comments

None.

Cite this review as

---

## Round 0.2 · Minor Revisions

Dear authors,

Thanks a lot for your efforts to improve the manuscript.

Nevertheless, some concerns are still remaining that need to be addressed.
You are advised to critically respond to the remaining comments point by point when preparing a new version of the manuscript and while preparing for the rebuttal letter.

Kind regards,
PCoelho

Reviewer 1 ·

Basic reporting

It is suggested that Figure 2 be improved to highlight the innovation of this article.

It is recommended to merge Section 6 and Section 7.

Experimental design

In the Abstract, this paper summarizes the shortcomings of the current GPS spoofing detection methods based on machine learning, one of which is "the accuracy of attack detection in resource-constrained environments for deployment". However, the method proposed in this paper is only simulated on PC, which can not verify that the method has solved this problem.

The detection of GPS spoofing attacks of UAV is a research hotspot. In order to make the method more comprehensive, it is necessary to compare the experimental data with more related methods.

Validity of the findings

How to solve the problem of data set imbalance in this paper?

Cite this review as

Reviewer 2 ·

Basic reporting

Thank you for response. I believe all the comments are answered and the paper is ready for publication.

Experimental design

The design and experiments are good and the comments are answered.

Validity of the findings

All good

Cite this review as

---

## Round 0.3 · Minor Revisions

Dear authors,

There is an inconsistency identified by the reviewer, please check: "Table 12 shows that Wei et al. (2022a) used simulation data. However, Table 4 shows that Wei et al. (2022a) used real data. Table 12 should be revised."

Please address all comments/suggestions provided by reviewers, considering that these should be added to the new version of the manuscript.

Kind regards,
PCoelho

Reviewer 1 ·

Basic reporting

The paper is good.

Experimental design

Table 12 shows that Wei et al. (2022a) used simulation data. However, Table 4 shows that Wei et al. (2022a) used real data. Table 12 should be revised.

Validity of the findings

All good.

Cite this review as

---

## Round 0.4 · accepted · Accept

Dear authors, we are pleased to verify that you meet the reviewer's valuable feedback to improve your research.

Thank you for considering PeerJ Computer Science and submitting your work.

Kind regards
PCoelho